# Clinical Validity of Anti-Proteinase 3 Antibodies in Patients with Inflammatory Bowel Disease: A Short Meta-Analysis

**DOI:** 10.3390/diagnostics13243682

**Published:** 2023-12-16

**Authors:** Carmen Andalucía, Laura Martínez-Prat, Chelsea Bentow, Mary Ann Aure, Michael P. Horn, Michael Mahler

**Affiliations:** 1Research and Development, Headquarters & Technology Center Autoimmunity, Werfen, San Diego, CA 92121, USA; candalucia@werfen.com (C.A.); lmartinez3@werfen.com (L.M.-P.); cbentow@werfen.com (C.B.); maure@werfen.com (M.A.A.); 2Department of Clinical Chemistry, Inselspital, Bern University Hospital, University of Bern, 3010 Bern, Switzerland; michael.horn@insel.ch

**Keywords:** PR3-ANCA, anti-PR3 antibodies, inflammatory bowel disease, IBD, ulcerative colitis, UC, Crohn’s disease, CD

## Abstract

Anti-neutrophil cytoplasmic antibodies (ANCA) directed to proteinase 3 (PR3) represent highly established markers for patients with ANCA-associated vasculitis (AAV). PR3-ANCA have also demonstrated utility in the management of inflammatory bowel disease (IBD). More specifically, PR3-ANCA discriminate individuals with ulcerative colitis (UC) from Crohn’s disease (CD) patients and are associated with disease severity, activity, and treatment non-response. Here, we aim to summarize the current data on the diagnostic utility of PR3-ANCA in IBD. A structured, systematic literature review, including three electronic databases, was conducted on June 6th, 2023, to identify studies assessing the diagnostic accuracy of the QUANTA Flash^®^ PR3 assay in UC vs. CD patients. Electronic searches were supplemented by hand searching. A hierarchical, bivariate, mixed-effect meta-analysis was conducted using the metandi function, as per the Cochrane collaboration recommendations. Study quality was assessed using the QUADAS-2 tool, which considers the risk of bias and applicability. Six out of a hundred and eleven citations met the inclusion criteria and reported QUANTA Flash^®^ PR3 diagnostic accuracy in UC vs. CD (UC, *n* = 667, CD, *n* = 682 patients). The sensitivity/specificity point estimate for UC was 34.9%/95.9%. This resulted in a Diagnostic Odds Ratio (DOR) of 12.6. The risk of bias was low in the index test and reference standard domains. Four of the six studies (67%) showed an unclear risk of bias in patient selection and in flow and timing domains. All studies had low concerns about applicability in all the domains. PR3-ANCA measured with the QUANTA Flash^®^ PR3 assay represent novel diagnostic markers in IBD and enables discrimination between UC and CD.

## 1. Introduction

Inflammatory bowel disease (IBD) is a chronic medical condition characterized by inflammation of the digestive tract. It comprises two main types: Crohn’s disease (CD) and ulcerative colitis (UC). CD can affect any portion of the digestive tract, from the oral mucosa to the anus, where lesions have a characteristic, discontinuous distribution and damage all the wall layers in depth. On the contrary, UC only affects the large intestine, where lesions can involve a variable but continuous proportion of the colon length, always starting in the rectum and only affecting the mucosal layer. IBD results from an abnormal immune response, where the immune system mistakenly attacks the gastrointestinal lining. Symptoms include abdominal pain, diarrhea, rectal bleeding, fatigue, and weight loss. These can vary in severity and may lead to complications like bowel obstructions, fistulas, and nutritional deficiencies. IBD’s exact etiology is unknown, but it is believed to involve genetic, environmental, and immune factors [1]. Disease management often involves medication, including treatment with biologicals, dietary changes, and, in severe cases, surgery. IBD is a lifelong condition that requires ongoing medical care and monitoring. Despite the recent advancements, several unmet needs in the diagnosis, management, and treatment of these patients remain, and biomarkers are gaining increasing attention for risk and patient stratification, to predict disease evolution, and for the development and implementation of treat-to-target strategies with the objective of overcoming the therapeutic ceiling being faced in IBD. 

Anti-neutrophil cytoplasmic antibodies (ANCA) are autoantibodies that target granule proteins in neutrophils, with myeloperoxidase (MPO) and proteinase 3 (PR3) representing two of the most relevant autoantigens. ANCA directed to PR3 are highly established diagnostic markers for patients with ANCA-associated vasculitis (AAV) and is particularly associated with granulomatosis with polyangiitis (GPA) [2]. 

Neutrophils are known to play an important role in IBD gut inflammation, and there is an increasing body of evidence indicating that ANCA may play a role in IBD pathology mediated by Neutrophil Extracellular Traps (NETs) [3]. Since the first report of PR3-ANCA in IBD in 1989 [4], several studies have reported this biomarker in a significant percentage of IBD patients, both adults and children, and clinical utility for the management of IBD has been proposed. Historically, indirect immunofluorescence (IIF) has played a key role in the screening of ANCA, and nowadays, antigen-specific immunoassays of high quality are available and, in many countries, generally used as the primary testing method [5,6]. These assays can be based on different technologies, including enzyme-linked immunosorbent assays (ELISA), chemiluminescence immunoassays (CIA), or multi-plex assays. 

The QUANTA Flash^®^ PR3 CIA has been reported to provide clinical validity in IBD diagnosis and classification [7]. More specifically, PR3-ANCA measured with this assay discriminated individuals with UC from CD patients and were associated with disease severity and activity [7,8,9,10]. Lastly, evidence is mounting that PR3-ANCA help stratify patients according to treatment response. 

In this study, we aimed to summarize the current data on the diagnostic utility of PR3-ANCA in IBD, with a special focus on the performance of the QUANTA Flash^®^ PR3 assay (Inova Diagnostics, San Diego, CA, USA).

## 2. Materials and Methods

Search strategy: A structured, systematic literature review including three electronic databases (Pubmed, Scopus, and Web of Science) was conducted on June 6th, 2023, using keywords (synonyms and acronyms) and MeSH terms for the target disease and comparator disease (“inflammatory bowel disease”, “Crohn’s disease”, and “ulcerative colitis”) and for the index test (“anti-PR3 antibodies”). Restrictions were not applied to any of the searches. Electronic searches were supplemented by hand searching reference lists of the included studies. Detailed search strategies are available in Appendix A.

Study selection and eligibility criteria: Records identified from electronic databases were imported into a citation manager (EndNote 20) for screening by two independent reviewers (CA and MM). The disagreement was resolved by consensus. After removing duplicates automatically (EndNote) and then manually, titles and abstracts were screened. Full-text articles with eligible citations were retrieved. Fully paired cohort or cross-sectional studies reporting QUANTA^®^ Flash PR3 diagnostic test accuracy in adults or children with a diagnosis of UC and CD were considered eligible. 

Data extraction: The first author´s name, publication year, region, index test, commercial test name, test method, test threshold, population type (adults/children), number of UC and CD, number of true positives, number of true negatives, number of false positives, number of false negatives, study design, and study setting were extracted by two independent authors in an Excel sheet.

Synthesis method: A hierarchical, bivariate, mixed-effect meta-analysis was conducted using the metandi function in STATA MP v17.0, as per the Cochrane collaboration recommendations for a review of diagnostic test accuracy studies. Exploratory analysis of heterogeneity was assessed by a visual examination of the hierarchical summary receiver operating characteristic (HSROC) curve and the forest plot created using metandi and midas functions, respectively, in STATA MP v17.0. A Fagan´s nomogram [11] was generated to graphically display an estimation of the post-test probability of UC in an individual patient given a certain pre-test probability. 

Quality assessment: The quality of the included studies was assessed using the QUADAS-2 tool, which considers the risk of bias and applicability. Signaling questions were tailored to facilitate the judgment of this review. Patient selection, the index test, the reference standard, and the timing and flow domains are part of the risk of biased evaluation. Applicability assessment is conducted on the first three domains. Each domain is rated as high, low, or unclear. Patient selection was rated as low risk of bias if the study avoided inappropriate exclusions, the cohort was clearly described, and it included samples at the time of diagnosis. Index test domain was rated as low risk of bias if a threshold was pre-specified and, therefore, a subjective interpretation of results was avoided. The reference standard domain was rated as having a low risk of bias if the study used international classification criteria for the target and comparator diseases, and patients were classified without knowledge of the result of the index test. The flow and timing domain was rated as having a low risk of bias if the index test was performed using samples at the time of diagnosis, all patients received the same reference standard, all patients received the same index test, and all patients were included in the analysis.

## 3. Results

A total of 150 records were identified through electronic and manual searches. After the removal of duplicates, 111 citations were title- and abstract-screened, and 17 full texts were reviewed. For the meta-analysis, we focused on studies based on the QUANTA Flash PR3, for which six studies met the inclusion criteria [7,8,9,12,13,14] (Figure 1). 

The six studies included a total of 1349 individuals, with 667 UC and 682 CD patients. Of them, five studies were based in Europe and one in Asia [8]. Two articles [12,13] were case-control studies including 369 children, of whom 154 had UC (46% males) and 215 had CD (61% males). Among individual studies, sensitivity varied from 18.4–41.7% in the adult population and from 40.5–57.6% in the child population, and specificity ranged from 93.6–98.9% in adults and from 89.3–92.9% in children (Table 1).

The QUANTA Flash^®^ PR3 showed a sensitivity point estimate of 34.9% [95% confidence interval (CI): 26.2–44.6%], a specificity point estimate of 95.9% (95% CI: 92.0–98.0%), a Diagnostic Odds Ratio point estimate (DOR) of 12.6 (95% CI: 5.7–27.9), a positive likelihood ratio (LR) point estimate of 8.5 (95% CI: 4.2–17.5), and a negative LR point estimate of 0.68 (95% CI: 0.59–0.78). The 95% CI (Figure 2) and the large 95% prediction region in the HSROC (Figure 3), particularly for sensitivity, indicate a moderate/high level of uncertainty in the estimates and should be taken with caution. A meta-regression analysis was not feasible due to the low number of studies.

From the clinical application point of view, given a patient with a pre-test probability of 25%, the post-test probability increased to approximately 74% when the QUANTA Flash^®^ PR3 was positive and decreased to approximately 18% in the case of a negative result (Figure 4).

The risk of bias was low in the index test and reference domains for all the included studies. All studies used international classification criteria as reference standards. In the patient selection, flow, and timing domains, four out of the six studies (67%) showed an unclear risk of bias based on the lack of information, and two studies were rated as low risk of bias based on the inclusion of all samples at the time of diagnosis. All studies had low concerns about applicability in all the domains (Figure 5).

As part of the literature review, we identified eleven studies that measured PR3-ANCA in patients with IBD using bead-based chemiluminescence assays. While most studies investigated the utility of PR3-ANCA to differentiate UC from CD, others looked at disease severity, activity, extent, and duration, as well as treatment response. A summary is provided in Table 2.

Two of the studies [10,15] found a significant difference in clinical activity measured by the partial Mayo score between PR3-ANCA-positive and PR3-ANCA-negative groups and an association between PR3-ANCA levels and clinical activity score excerpts between moderate and severe (*p* > 0.05). In contrast, another study [14] showed that patients with severe disease presented higher PR3-ANCA levels than individuals with moderate disease (*p* < 0.05).

Endoscopic mucosal injury classified by Mayo endoscopy subscore (MES) was examined in three of the studies [14,17,18]. PR3-ANCA-positive UC patients had significantly higher MES compared to the PR3-ANCA-negative group [17]. Moreover, there was a significant positive correlation between PR3-ANCA levels and MES [17]. In addition, PR3-ANCA positivity demonstrated to be the best laboratory parameter in predicting endoscopic activity and was correlated with the duration of hospital stay [18]. PR3-ANCA levels were also positively correlated with the pathological activity assessed by Matts grade [17].

Four studies [8,10,14] investigated differences in colitis extension based on the Montreal classification or a pediatric modification of the Montreal classification [13] and PR3-ANCA levels or positivity. PR3-ANCA positivity was significantly more prevalent in individuals presenting pancolitis than proctosigmoiditis, or left-sided disease. PR3-ANCA levels also progressively increased with the extension of the disease.

While a study reported a significantly shorter disease duration in the PR3-ANCA-positive group compared to the PR3-ANCA-negative group [15], a multivariate analysis [8] did not find this association independently.

Finally, multivariate analyses showed that PR3-ANCA positivity was statistically associated with primary nonresponse to anti-TNF-α agents (odds ratio: 19.3, 95% CI: 3.3–172.7, *p* = 0.002) [16] and to steroid treatment (odds ratio: 5.19, 95% CI: 1.5–17.5, *p* = 0.008) [17].

## 4. Discussion

Our research includes a systematic literature review and a meta-analysis on the utility of PR3-ANCA, measured by a bead-based chemiluminescence assay, to distinguish patients with UC and CD. The short meta-analysis showed that PR3-ANCA, measured by the QUANTA Flash PR3, are detected in 34.9% of UC patients and in 4.1% of individuals with CD. Although our meta-analysis was focused on a specific PR3-ANCA assay to avoid heterogeneity associated with technology differences, we found a large 95% CI in sensitivity, which could be explained by the cohort. Indeed, the heterogeneity in sensitivity observed in the child population was low. Further studies would be needed to investigate the large heterogeneity in sensitivity in the adult population. A descriptive analysis was included on the clinical utility of PR3-ANCA as a biomarker of disease severity, activity, extension or duration, and treatment response due to the limited number of studies evaluating these features. In UC patients, PR3-ANCA have shown a correlation with the partial Mayo score, Mayo endoscopy subscore, Matts grade, and Montreal classification. Moreover, a statistically significant association between PR3-ANCA and non-response to anti-TNF α agents and steroid treatment has recently been reported. The data summarized in this study holds promise for the clinical value of PR3-ANCA. However, further research is needed to understand any potential role of PR3-ANCA in the diagnosis, prognosis, and monitoring of IBD.

Several biomarkers have been developed and applied to aid in the diagnosis of IBD [20,21]. Among them, fecal calprotectin is commonly used as a rule-out test for IBD. In addition, the anti-Saccharomyces cerevisiae antibody (ASCA, both IgG and IgA) is used for the differentiation between CD and UC, being present in 20–30% of CD and 5–10% of UC patients. On the other hand, atypical pANCA frequently occur in patients with IBD, more specifically in patients with UC, with a sensitivity of about 35–50% and a specificity against CD of ~90% [2,19]. 

In addition to the biomarkers indicated above, most recently, IgG to integrin αvβ6 has been reported to be mostly found in UC compared to CD [22,23,24,25,26]. In the training and validation groups, 103/112 (92.0%) patients with UC and only 8/155 (5.2%) controls had anti-integrin αvβ6 antibodies (*p* < 0.001), resulting in a sensitivity of 92.0% and a specificity of 94.8% for diagnosing UC. Anti-integrin αvβ6 antibody titers correlated with UC disease activity, and IgG1 was the major subclass. Patient IgG is bound to the integrin αvβ6 expressed on colonic epithelial cells. Moreover, IgG in patients with UC blocked integrin αvβ6-fibronectin binding through an RGD (Arg-Gly-Asp) tripeptide motif and inhibited cell adhesion.

### 4.1. PR3-ANCA as an IBD Diagnostic Marker

The first report of PR3-ANCA in IBD dates back more than three decades, and since then, several studies have confirmed the presence of this marker in the serum of a significant percentage of IBD patients. However, to date, very limited reports have systematically investigated the clinical utility of this biomarker in IBD. In 2020, a literature review examined the correlation between PR3-ANCA and different IBD clinical characteristics [14] and confirmed that this biomarker could represent an important tool to support IBD diagnosis in the discrimination between UC and CD and for prognosis (disease extent and severity). Unfortunately, most of these studies were performed on patients with a Caucasian background, with a few on Asian populations. Given that the incidence of IBD is growing worldwide and especially in minority groups [27], and to advance equity of care in the diagnosis and management of IBD patients, it will be key to ensure representation of patients from different ethnicities in future studies. 

On the other hand, technical and performance differences between technologies and assays for the detection of PR3-ANCA in IBD have been reported [28,29,30,31]. CIA has shown superior performance to IIF (atypical pANCA), which is associated with a significantly higher specificity of CIA [8]. A recent study compared a fluorometric ELISA (FEIA) and two bead-based assays, a CIA and a multiplex assay, for the detection of anti-PR3 antibodies and assessed their ability to differentiate AAV from controls as well as UC from CD. The results showed that while the performance of the three assays for AAV was comparable, significant differences were observed for the application in IBD, and in this study, the bead-based assays showed better discrimination compared to the micro-well-based system [7]. 

These differences in assay performance in IBD remain somewhat illusive. In addition to the intrinsic technical differences between technologies, the PR3 antigen and its exposure to the solid phase represent key factors. The majority of anti-PR3 antigen-specific assays use human native PR3 protein as an antigen; however, some assays utilize mixtures of human native and recombinant proteins. On the other hand, compared to other platforms, the negative charge of the microparticles used in the QUANTA Flash^®^ PR3 CIA could favor the exposure of the clinically relevant PR3 epitopes in IBD. Although this is speculative, there is evidence that PR3 epitopes in AAV might play a pathological role [32]. 

Due to these differences in diagnostic performance between platforms, we decided to focus on the most commonly used PR3-ANCA assay in IBD studies. Our meta-analysis of the diagnostic utility of PR3-ANCA measured using the QUANTA Flash PR3 CIA method demonstrated high levels of consistency among patient cohorts. A total of 667 UC patients and 682 individuals with CD were included in the analysis, leading to a pooled sensitivity point estimate of 34.9% [95% CI: 26.2–44.6%], a specificity point estimate of 95.9% (95% CI: 92.0–98.0%), a diagnostic odds ratio point estimate (DOR) of 12.6 (95% CI: 5.7–27.9), a positive likelihood ratio (LR) point estimate of 8.5 (95% CI: 4.2–17.5), and a negative LR point estimate of 0.68 (95% CI: 0.59–0.78). These results confirm the utility of anti-PR3 antibodies measured with this assay to discriminate between UC and CD and to help improve IBD diagnosis. 

### 4.2. Combinations of Biomarkers

Combining PR3-ANCA with other serological markers has the potential to further improve the differential diagnosis of UC and CD [12,19]. On the other hand, these combinations so far have not been able to classify IBD unclassified (IBD-U) patients, neither to one of the established entities nor to predict the reclassification of these individuals. Along those lines, a scoring system has recently been developed to predict the severity of UC based on a combination of different markers, including PR3-ANCA [18]. Until today, no study has been conducted to test antibodies against integrin αvβ6 and PR3-ANCA in the same cohort. Based on the promising findings with individual markers and the fast-growing field of machine learning, such studies are highly anticipated. 

### 4.3. Pathogenic Role of Anti-PR3 Antibodies

While the understanding of the role of PR3-ANCA in AAV pathogenesis has significantly advanced over the last few years, little is known in the context of IBD. The most relevant factors influencing PR3-ANCA pathogenicity in GPA are related to their interaction with neutrophils: the level of PR3 autoantigen at the neutrophil surface, the epitope of PR3 recognized by PR3-ANCA, isotype, and glycosylation of PR3-ANCA [32]. 

Anti-PR3 antibodies could represent a connecting point between AAVs and IBD, and they could help explain the clinical overlap between these two conditions [29]. Several case reports have been published hinting at such an interface [33].

In IBD, it is well established that neutrophils play an important role in gut inflammation, and recent data has indicated that ANCA may play a role in IBD pathology mediated by NETs. He et al. showed that treatment of neutrophils with isolated IgG from PR3-ANCA-positive active IBD patients resulted in the release of NETs [34]. More recently, another study demonstrated that NETs exacerbate colon tissue damage and drive thrombotic tendency during active IBD and proposed a mechanism in which ANCA might decrease the breakdown of NETs by attenuating DNase I activity [35]. 

In any event, whether anti-PR3 antibodies have a direct pathogenic role in IBD and their connection with NET formation and neutrophil activation remains to be elucidated, and studies with this research goal are warranted, especially in the context of novel therapeutic strategies and patient stratification for treat-to-target. 

### 4.4. Disease Activity and Prediction of Treatment Response

PR3-ANCA could also represent a promising non-invasive biomarker in UC for assessment of disease activity, typically measured with scoring systems such as the Mayo score or the UC Endoscopic Index of Severity (UCEIS) [10,17], and of disease extent, which is often used as an indicator of disease severity [8,10,14]. Some of the early studies that showed this association with disease extent and severity were conducted in the UK [8] and China [14] with the QUANTA Flash PR3 assay, and more recently, additional research performed in China and Japan has validated these findings in additional cohorts [10,14,18]. 

In addition to this proposed prognostic value, several interesting studies have recently indicated that PR3-ANCA could be useful for the prediction of clinical courses in UC [10,16,17,36,37]. Mayo scores and PR3-ANCA titers seem to significantly decrease with treatment, and significant reductions of PR3-ANCA titers have been observed in patients in clinical remission. On the other hand, published data indicate that this marker is associated with failure of response to steroid therapy [17] and that it could be a predictor of primary non-response to anti-tumor necrosis factor (TNF) agents [16], with an odds ratio of 19.3 (*p* = 0.002). These studies were all performed in Japan and therefore warrant replication in global cohorts. Since TNF inhibitors are widely used in patients with IBD worldwide, predicting non-response to these therapeutic agents could help solve a critical unmet need. 

## 5. Conclusions

In this meta-analysis, the QUANTA Flash^®^ PR3 assay has proved to be a useful autoantibody to discriminate between UC and CD with a sensitivity of 34.9% and a specificity of 95.9%. In addition to this utility in IBD diagnosis, PR3-ANCA represents a promising biomarker for disease prognosis, monitoring, and treatment non-response prediction in patients suffering from IBD. Further research is needed to validate these applications, and clinical trials should consider the inclusion of PR3-ANCA as an exploratory biomarker for patient stratification. 

## Figures and Tables

**Figure 1 diagnostics-13-03682-f001:**
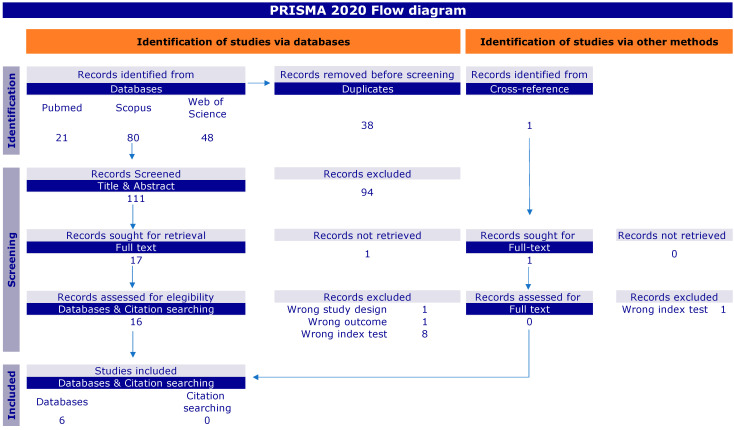
PRISMA 2020 flow diagram for new systematic reviews, which includes searches of databases, registers, and other sources.

**Figure 2 diagnostics-13-03682-f002:**
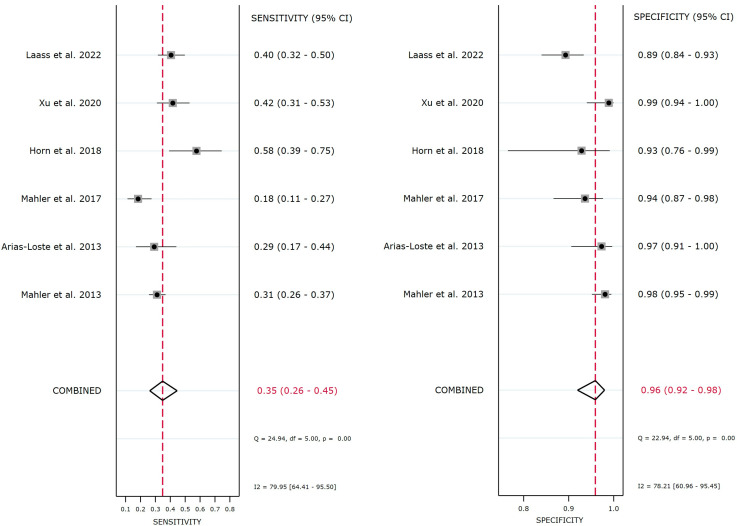
Forest plot of sensitivity and specificity of the PR3-ANCA for detection of ulcerative colitis (UC) and Crohn’s disease (CD). The vertical red dotted line represents the meta-analysis summary estimate, which corresponds to the value of 0.35 for sensitivity and 0.96 for specificity. A total of six studies are shown individually and combined for sensitivity and specificity. The combined sensitivity and specificity were 34.9% and 95.9%, respectively [7,8,9,12,13,14].

**Figure 3 diagnostics-13-03682-f003:**
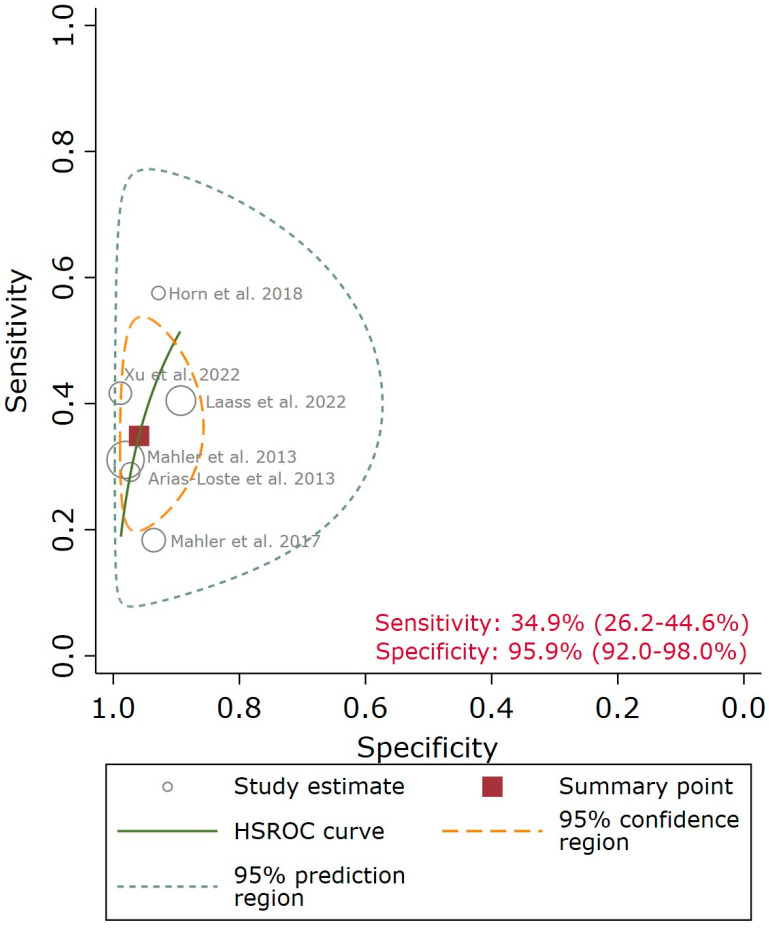
Summary receiver operating characteristic plot of PR3-ANA to discriminate between ulcerative colitis (UC) and Crohn’s disease (CD). The size of each point is scaled according to the precision, sensitivity, and specificity of the study [7,8,9,12,13,14]. The solid circle (summary point) represents the summary estimate of sensitivity and specificity for PR3-ANCA. The summary point is surrounded by a dotted line representing the 95% confidence region and a dashed line representing the 95% prediction region (the region within which we are 95% certain that the results of a new study will lie).

**Figure 4 diagnostics-13-03682-f004:**
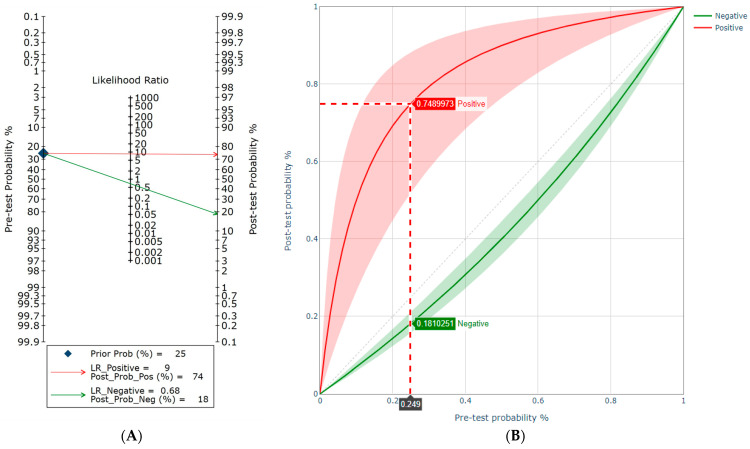
Impact of the PR3-ANCA test result on the probability of ulcerative colitis (UC). (**A**) Fagan´s nomogram for the QUANTA Flash^®^ PR3 assay. For example, the QUANTA Flash PR3 assay in high-risk patients has an estimated diagnostic sensitivity (DSe) of 34.9% and specificity (DSp) of 95.9% for ulcerative colitis (UC). It is necessary to calculate first the likelihood ratio of positive and negative test results separately (LR+ and LR−, respectively) using conventional formulas (LR+ =  DSe/(1 − DSp) and LR−  =  (1 − DSe)/DSp). Given that the patient came from a high-risk population with an estimated prevalence of 25%, if this patient tests positive, the post-test probability that this person truly has UC would be approximately % (red line). Alternatively, if the patient tests negative, the post-test probability that the patient truly has UC would be approximately 18% (green line). (**B**) Pre-test/post-test probability plots.

**Figure 5 diagnostics-13-03682-f005:**
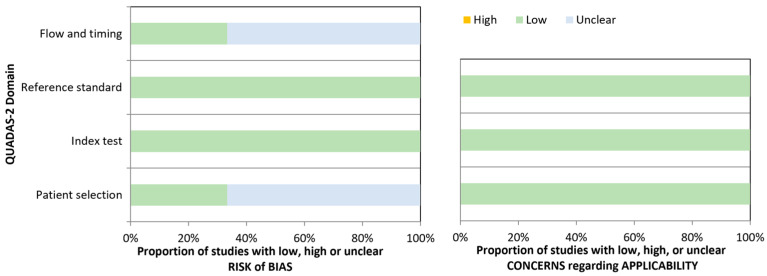
Summary of the Quality Assessment of Diagnostic Accuracy Studies (QUADAS-2): evaluating the risk of bias and applicability concerns. It consists of four domains (patient selection, index test, reference standard, flow, and timing). All domains are rated as “low”, “high”, or “unclear” in relation to the risk of bias, and the first three domains in regard to applicability concerns.

**Table 1 diagnostics-13-03682-t001:** Summary of the diagnostic test accuracy of the individual studies.

Study	UC PR3-ANCA Pos	UC Total	Sensitivity (95% CI)	CD PR3-ANCA Pos	CD Total	Specificity (95% CI)
Laass et al., 2022 [13]	49	121	40.5% (31.7–49.8%)	20	187	89.3% (84.0–93.3%)
Xu et al., 2022 [14]	35	84	41.7% (31.0–52.9%)	1	91	98.9% (94.0–100%)
Horn et al., 2018 [12]	19	33	57.6% (39.2–74.5%)	2	28	92.9% (76.5–99.1%)
Mahler et al., 2017 [7]	18	98	18.4% (11.3–27.5%)	6	94	93.6% (86.6–97.6%)
Mahler et al., 2013 [8]	88	283	31.1% (25.9–36.8%)	4	208	98.1% (95.2–99.7%)
Arias-Loste et al., 2013 [9]	14	48	29.2% (17.0–44.1%)	2	74	97.3% (90.6–99.7%)
Estimates	223	667	34.9% (26.2–44.6%)	35	682	95.9% (92.0–98.0%)

Abbreviations: ANCA = anti-neutrophil cytoplasmic antibodies; CI = confidence interval; CD = Crohn’s diseases; Pos = positive; PR3 = proteinase 3; UC = ulcerative colitis; CI = confidence interval.

**Table 2 diagnostics-13-03682-t002:** Overview of studies on PR3-ANCA on inflammatory bowel disease (IBD) using bead-based assays.

	Assay	UC vs. CD (AUC)	Disease				Treatment Response	
			Severity	Activity (MES)	Extent	Duration	Anti-TNFi	Steroid
Arias-Loste et al., 2013 [9]	QF	X 0.81						
Mahler et al., 2013 [8]	QF/QL	X 0.76/0.60	x		x	x		
Takedatsu et al., 2018 [15]	STACIA	X 0.85	x			x		
Horn et al., 2018 * [12]	QF	N/A						
Xu et al., 2020 [14]	QF	X 0.89	x	x	x			
Imakiire et al., 2021 [10]	STACIA	X 0.76	x		x			
Yoshida et al., 2021 [16]	STACIA						x	
Aoyama et al., 2021 [17]	STACIA			x				x
Laass et al., 2022 * [13]	QF	X 0.74			x			
Zeng et al., 2023 [18]			x	x				
Sokollik et al., 2023 [19]	QF	N/A						

TNFi = tumor necrosis factor inhibitor; QF = QUANTA Flash; QL = QUANTA Lite; N/A = not applicable. * pediatric population.

## Data Availability

Data can be made available on request.

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
