# Peer review of "Clinical Validity of Anti-Proteinase 3 Antibodies in Patients with Inflammatory Bowel Disease: A Short Meta-Analysis"

_diagnostics, 2023, doi:10.3390/diagnostics13243682_

Round 1

Reviewer 1 Report

Comments and Suggestions for Authors

The manuscript entitled "Clinical validity of anti-proteinase 3 antibodies in patients with 2 inflammatory bowel disease: a systematic review and meta- analysis" is an interesting work - A short meta analysis including three electronic databases. 

Since it is not a structured systematic literature review as written in the title and abstract, this should be omitted. The title could be: "Clinical validity of anti-proteinase 3 antibodies in patients with 2 inflammatory bowel disease: a short meta- analysis"

Kindly check if the abstract is written in accordance with instructions for authors.

Figures 2, 3 and 4 are unreadable, font should be adjusted. 

The discussion section is insufficient and important findings should be discussed in more detail.

Comments on the Quality of English Language

English language needs minor polishing.

Reviewer 2 Report

Comments and Suggestions for Authors

Using meta-analysis, the authors seek to answer whether PR3-ANCA can be a suitable biomarker to distinguish UC from CD.
The structure of the meta-analysis is appropriate and follows the PRISMA 2020 guideline.
The selection of articles included in the study is also appropriate.
Statistical methods were selected according to the study objective.
The results are clear; the figures and illustrations are illustrative and help to understand the content of the article.

However, I would have questions about the Discussion:
How does your result differ from the previously reported result?
Imakiire, S.; Takedatsu, H.; Mitsuyama, K.; Sakisaka, H.; Tsuruta, K.; Morita, M.; Kuno, N.; Abe, K.; Funakoshi, S.; Ishibashi, H.; et al. Role of Serum Proteinase 3 Antineutrophil Cytoplasmic Antibodies in the Diagnosis, Evaluation of Disease Severity, and Clinical Course of Ulcerative Colitis. Gut Liver 2022, 16, 92-100.

Is it really possible that ANCA testing will play an important role in the diagnosis of IBD in the future? Is it even conceivable that the test will replace the role of endoscopy and histology?
What would be an appropriate recommendation for cases where clinical and histological findings do not allow us to distinguish UC from CD?
I suggest considering the markers listed in the following article.
D'Incà R, Sturniolo G. Biomarkers in IBD: What to Utilize for the Diagnosis?. Diagnostics (Basel). 2023;13(18):2931. Published 2023 Sep 13. doi:10.3390/diagnostics13182931

ANCA antibodies have different epitope specificities, so not all ANCA are pathogenic. Is there data on the proportion of non-specific ANCA types (i.e., x-ANCA or aspecific ANCA) in IBD?

In the international literature, Crohn's disease is mostly abbreviated as CD. I suggest that authors also use the abbreviation CD instead of CrD.

I recommend that the above points be discussed in the Discussion.

A major revision is required.

Round 2

Reviewer 1 Report

Comments and Suggestions for Authors

Authors improved the manuscript.

Comments on the Quality of English Language

English needs moderate revision.

Reviewer 2 Report

Comments and Suggestions for Authors

The authors have revised the manuscript correctly. The revised version is now acceptable for publication.